# Genomic Analysis of *Stropharia rugosoannulata* Reveals Its Nutritional Strategy and Application Potential in Bioremediation

**DOI:** 10.3390/jof8020162

**Published:** 2022-02-06

**Authors:** Ying Yang, Guoliang Meng, Shujun Ni, Haifeng Zhang, Caihong Dong

**Affiliations:** 1State Key Laboratory of Mycology, Institute of Microbiology, Chinese Academy of Sciences, Beijing 100101, China; yangy@im.ac.cn (Y.Y.); 18706387822@163.com (G.M.); 2Institute of Animal Husbandry Research, Heilongjiang Academy of Agricultural Sciences, Harbin 150086, China; nishujun66@163.com (S.N.); hfzhang0000@163.com (H.Z.)

**Keywords:** *Stropharia rugosoannulata*, lignin and xenobiotic, peroxidase, litter-decomposing fungi, siderophore, bioremediation

## Abstract

*Stropharia rugosoannulata* is not only a popular edible mushroom, but also has excellent potential in bioremediation. In this study, we present a high-quality genome of a monokaryotic strain of the *S. rugosoannulata* commercial cultivar in China. The assembly yielded an N50 length of 2.96 Mb and a total size of approximately 48.33 Mb, encoding 11,750 proteins. The number of heme peroxidase-encoding genes in the genome of *S. rugosoannulata* was twice the average of all of the tested Agaricales. The genes encoding lignin and xenobiotic degradation enzymes accounted for more than half of the genes encoding plant cell wall degradation enzymes. The expansion of genes encoding lignin and xenobiotic degradation enzymes, and cytochrome P450 involved in the xenobiotic metabolism, were responsible for its strong bioremediation and lignin degradation abilities. *S. rugosoannulata* was classified as a litter-decomposing (LD) fungus, based on the analysis of the cell wall degrading enzymes. Substrate selection for fruiting body cultivation should consider both the nutritional strategy of LD and a strong lignin degradation ability. Consistent with safe usage as an edible mushroom, the *S. rugosoannulata* genome does not contain genes for known psilocybin biosynthesis. Genome analysis will be helpful for understanding its nutritional strategy to guide fruiting body cultivation and for providing insight into its application in bioremediation.

## 1. Introduction

*Stropharia rugosoannulata* Farl. Ex Murrill, called saketsubatake in Japanese and wine-cap *Stropharia* in English, is a popular edible mushroom with high nutritional and medicinal values. *S. rugosoannulata* was first domesticated in Germany in the 1960s [1] and was recommended by the Food and Agriculture Organization as a cultivated mushroom for developing countries [2]. By 1989, commercial production of *Stropharia* in Europe reached approximately 1300 tons per year [3]. Field cultivation of *S. rugosoannulata* has expanded in most of the provinces in China, and the cultivation area was estimated to be approximately 1333 ha in 2019.

In recent years, the consumption of *S. rugosoannulata* has increased dramatically due to its excellent taste and pharmacological activities. In addition to its nutritional and medicinal values [4,5], *S. rugosoannulata* has great potential for bioremediation. It can degrade a wide range of structurally different environmental pollutants, including polycyclic aromatic hydrocarbons [6], synthetic dyes [7], 2,4,6-trinitrotoluene [8], bisphenol A [9], and dibenzo-p-dioxins and dibenzofurans [10], as well as pharmaceutical compounds such as carbamazepine, venlafaxine, iopromide, diclofenac, cyclophosphamide, and ifosfamide [11]. *S. rugosoannulata* has been found to be a promising fungal species for pharmaceutical biodegradation in contaminated water [11]. It has been confirmed that *S. rugosoannulata* can degrade diuron, a stable herbicide that may pose a severe threat to human health and aquatic organisms in the environment [12].

Various agricultural byproducts have been used as substrates for the fruiting body cultivation of this fungus. In 1978, Szudyga chose a substrate composed of cereal or flax straw to cultivate *S. rugosoannulata* and to produce fruiting bodies [1]. Bruhn et al. [13] cultivated *S. rugosoannulata* with two substrate systems: an uncased mixed wood chip/soil substrate, and a straw substrate cased with the wood chip/soil mixture. Li et al. [14] cultivated *S. rugosoannulata* in a small arch shed with mulberry sawdust. Gong et al. [15] selected a mixture of rice husk, corncobs, and sawdust to cultivate *S.*
*rugosoannulata*. In this laboratory, we used bagasse fiber and straw as the growth substrates. The pruning wood chips of *Tilia* and *Populus* and *Alnus* support greater *Stropharia* yields than straw [3]. Sawdust generally accounts for 30% of the cultivation substrate in China. Analysis of the enzymes involved in substrate degradation will be helpful for the optimization of cultivation substrates and high-quality cultivation.

From an ecophysiological point of view, saprotrophic basidiomycetes can be classified into three partially overlapping groups, namely, brown rot (BR), white rot (WR), and litter-decomposing (LD) fungi. *S. rugosoannulata* can assimilate both cellulosic straw and lignin sawdust; it is categorized as LD [9,16] or WR [17,18]. This is different from a typical LD, such as *Agaricus bispora**,* which is usually cultivated on wheat and rice straw. Enzymatic activity assays have revealed that *S. rugosoannulata* produces extracellular peroxidases during growth in beech wood microcosms and that manganese-oxidizing peroxidases are the predominant enzymes of lignin degradation [19]. Defining the nutritional strategy of *S. rugosoannulata* will be helpful for substrate selection during fruiting body cultivation. 

Recently, genome sequences of 90 fungal species, including *S. rugosoannulata*, were determined. A sample of a dikaryotic fruiting body of *S. rugosoannulata* was used, which could have affected the genome assembly [20]. The quality of the genomic data needs to be improved, since 17,452 scaffolds and an N50 of 8626 bp were obtained by Illumina sequencing. The complete mitochondrial genome sequence of *S. rugosoannulata* has been determined, and phylogenetic analysis based on the mitochondrial genome showed that *S. rugosoannulata* is closely related to *Coprinopsis cinerea* [21]. The structures of mating loci have been characterized by bioinformatics based on genomic data without any other genome information [22]. The mechanisms of thermotolerance [23] and cold resistance [24] have been revealed by transcriptome analysis in *S. rugosoannulata*; however, there is no reference genome of this species [23,24]. The high-quality genome of this fungus is needed for further research.

Genome sequencing of a few LD fungi has revealed variations in the genes encoding the plant cell wall degradation (PCWD) machinery among species [25]. In the present study, the high-quality genome of *S. rugosoannulata* was assembled based on a monokaryotic strain originating from the Chinese commercial cultivar “Heinong No. 1”. Comparative analyses revealed a versatile suite of biopolymer-degrading enzymes and the nutritional strategy. The genomic analysis will contribute greatly to fruiting body cultivation and application in the bioremediation of this species.

## 2. Materials and Methods

### 2.1. Origin of the Strain, Culture Conditions, and DNA/RNA Preparation

The strain “Heinong No. 1” (CGMCC 5.2220) is a culture of commercial cultivar of *S. rugosoannulata* in China. The monokaryotic strain was obtained as follows: First, the fruiting bodies of strain “Heinong No. 1” were cultivated, and basidiospores were collected. Then, the basidiospores were sprayed on potato dextrose agar (PDA) and incubated at 25 °C for 10 days until spore germination. Thereafter, single spore isolates without any clamp connection were selected under 600× magnification using an optical microscope (Eclipse 80i, Nikon, Tokyo, Japan) and were subcultured individually on PDA plates at 25 °C (Figure 1).

A vegetative monokaryotic strain S68 derived from the “Heinong No. 1” strain was cultured in potato dextrose broth (PDB) at 25 °C and 150 rpm for 7 days. Mycelia were harvested by filtering and being ground in liquid nitrogen. High-quality genomic DNA was then extracted using the QIAGEN^®^ Genomic kit (Qiagen, Dusseldorf, Germany) following the manufacturer’s instructions. Total RNA was extracted using TRIzol reagent (Invitrogen, Carlsbad, CA, USA).

### 2.2. Library Construction, Genome/Transcriptome Sequencing, and Assembly

The Illumina HiSeq X-Ten and PacBio SEQUEL platforms were used for genome sequencing at Nextomics Biosciences Co., Ltd. (Nextomics Biosciences, Wuhan, China). A 350-bp library was constructed following Illumina’s standard protocol, and it was subjected to paired-ended 150-bp sequencing by Illumina HiSeq X-ten. The sequencing data (clean bases: 6.52G, ~129X coverage of the estimated genome size; Appendix A) were used to estimate the genome size, repeat content, and heterozygosity.

Then, a 20-kb library was constructed following PacBio’s standard methods. The library was quantified using NanoDrop (Thermo Scientific, Waltham, MA, USA) and Qubit (Invitrogen, Carlsbad, CA, USA), and then sequenced through single-molecule real-time (SMRT) sequencing (PacBio Sequel, PacBio, Menlo Park, CA, USA). The sequencing data (filtered reads: 14.56 G, ~266X coverage of the estimated genome size; Appendix A) were assembled using Falcon (v1.8.1) with the default parameters. The completeness of the assembled genome was evaluated using the Core Eukaryotic Genes Mapping Approach (CEGMA) v2 and Benchmarking Universal Single-Copy Orthologs (BUSCO v 4.0.5) with conserved orthologous gene profiles for fungi.

For the transcriptome analysis, the strain “Heinong No. 1” was cultured on PDA media and the mycelia were collected for RNA extraction. Three libraries were generated using the NEB Next Ultra RNA Library Prep Kit for Illumina (NEB, Ipswich, MA, USA) and were sequenced on an Illumina HiSeq X-ten platform (Illumina Inc., San Diego, CA, USA) by Nextomics Biosciences Co., Ltd. (Wuhan, China).

### 2.3. Repeat Annotation, Gene Prediction, Gene Function, and Noncoding RNA Annotation

Tandem repeats were identified using GMATA V2.2 [26]. For the transposable element (TE), ab initio prediction was performed using RepeatModeler version open-1.0.11 to establish a de novo repeat sequence library, which was then classified by TEclass [27]. Finally, RepeatMasker Revision 1.331 was used to search against the de novo repeat sequence database generated from RepeatModeler’s prediction.

Gene models were constructed by combining ab initio, homology-based, and RNA-seq-assisted prediction. Augustus V3.3.1 was used for ab initio gene prediction. In the homology-based prediction, protein sequences from five fungi, *Hypholoma sublateritium* FD-334 SS-4 [28], *Agrocybe praecox* (https://mycocosm.jgi.doe.gov/Agrpra2/Agrpra2.home.html. 30 June 2020), *Pholiota highlandensis* (https://mycocosm.jgi.doe.gov/Phohig1/Phohig1.home.html. 30 June 2020), *Pholiota alnicola* (https://mycocosm.jgi.doe.gov/Phoaln1/Phoaln1.home.html. 30 June 2020), and *Psilocybe cubensis* [29], were used to construct exact gene models from all initially aligned coding sequences using GeMoMa v.1.6.2 with the default parameters. Additionally, the RNA-seq data obtained from mycelia were mapped to the genome assembly using Hisat2 v2.1.0 [30].

The predicted genes of *S. rugosoannulata* were annotated by alignment against the non-redundant Protein Database in National Center for Biotechnology Information (NCBI), Kyoto Encyclopedia of Gene and Genomes (KEGG), Eukaryotic Orthologous Groups of protein (KOG), Gene Ontology (GO), and SwissProt databases. Furthermore, motifs and domains were annotated using InterPro scan 5.32-71.0.

Non-coding RNAs, including rRNAs, snRNAs, microRNAs, and tRNAs, were identified by adopting infernal v1.1.2 using the Rfam database [31] for the *S. rugosoannulata* genome using BLASTN (E-value ≤ 1e^−5^). Transfer RNA was predicted using tRNAs can-SE v2.0 software with the default parameters for eukaryotes. The rRNAs and their subunits were predicted using RNAmmer v1.2. Protein targeting predictions were made using SignalP and TMHMM.

Furthermore, the genome completeness of the assembly was evaluated by BUSCO, RNA-seq data, and comparative analysis with the genomes of closely related genetic species.

### 2.4. Phylogenomic Analyses

Phylogenomic analyses were performed using 16 fungal genomes (Appendix A). Gene families with only one gene per species were identified by clustering protein sequences using OrthoMCL [32], resulting in 1609 single-copy orthologous genes of the 16 species. The amino acid sequences of 1609 single-copy genes were aligned using MAFFT version 7.0 (https://mafft.cbrc.jp/alignment/server/. 25 June 2020) with the default parameters, and poorly aligned sequences were then eliminated using Gblocks version 0.91b [33]. The conserved regions were concatenated into one sequence using Sequence Matrix-Windows 1.7.8 [34]. A maximum likelihood phylogenomic tree was generated using the concatenated amino acid sequences with RAxML version 8 [35] with the amino acid replacement matrix LG/WAG selected by ProtTest version 3.4.2 [36]. *Grifola frondosa* was used as an outgroup. One thousand bootstrap replicates were used, and the trees were shown in FigTree v1.4.2 [37]. Bootstrap values higher than 50% from RaxML were indicated in the phylogenetic trees.

### 2.5. Putative Peroxidase- and Carbohydrate-Active Enzyme (CAZyme)-Encoding Genes in the Genome of S. rugosoannulata

Putative peroxidase-encoding genes were identified from the fungal peroxidase database (fPoxDB; http://peroxidase.riceblast.snu.ac.kr. 20 October 2020) with three parameters: E-value < 1e^−5^, number of matching sequences ≥ 500, and matrix BLOSUM62 for BLASTP [38]. The CAZymes in the *S. rugosoannulata* genome were identified using dbCAN2 (http://bcb.unl.edu/dbCAN2/. 27 August 2020) with three tools: E-value < 1e^−15^ and coverage > 0.35 for HMMER+dbCAN; E-value < 1e^−102^ for DIAMOND+CAZy; a number of conserved peptide hits >6 and a sum of conserved peptide frequencies >2.6 for Hotpep+PPR [39]. Those with # of Tools ≥ 2 were kept as candidates.

### 2.6. Classification of the Nutritional Strategy of S. rugosoannulata by Linear Discriminant Analysis (LDA) Based on the PCWD Gene Families

LDA is used in statistics and other fields to find a linear combination of features that characterize or separate two or more classes of objects or events [40]. Forty-four PCWD gene families that were sorted into three functional categories, namely, cellulose, hemicellulose (including pectin), and lignin/xenobiotic degradation [25], were identified in the genome of *S. rugosoannulata* according to the Pfam, Interpro, and SSF identifiers. The nutritional strategy of *S. rugosoannulata* was determined by LDA with SPSS version 25.0 (IBM Corporation, Armonk, NY, USA) using the protein dataset for these 44 PCWD gene families of three different nutritional strategies (BR, WR, and LD) [25].

### 2.7. Prediction of the Secondary Metabolite Gene Clusters and Cytochrome P450- Encoding Genes

Secondary metabolite gene clusters were predicted with fungal AntiSMASH 3.0 (https://fungismash.secondarymetabolites.org/. 23 July 2020). Cytochrome P450 (CYP) proteins were predicted by aligning the gene models to the fungal P450 database (http://p450.riceblast.snu.ac.kr/index.php?a=view. 10 August /2020) using BLASTP (E-value < 1e^−10^, identity ≥ 40%, coverage ≥ 55%, and matrix = BLOSUM62). The CYP proteins were assigned to protein families based on Nelson’s nomenclature [41]. For protein sequences that were aligned with multiple families, the top hit was chosen.

## 3. Results

### 3.1. Sequencing Output Processing and De Novo Genome Assembly

The *S.*
*rugosoannulata* genome was sequenced by both the Illumina and PacBio SMRT platforms. Subread distribution analyses confirmed the high quality of the 20-kb library (Appendix A). There was no apparent heterozygous peak, and the heterozygosity was low at 0.01% (Appendix A). The resulting assembly yielded 48.33 Mb from ~266X coverage (Table 1), comprising 21 scaffolds with 100% of the genome assembled in the scaffolds exceeding 5 kb in length (Appendix A and Figure 2). Telomeres were identified manually by sequence observation, and all were TTTAGGG repeats of approximately 100 bp in length. Two contigs with telomeric repeats on both ends were found, and 14 of these scaffolds contained characteristic (TTTAGGG)n telomeric repeats on either the 5′ or 3′ end (Appendix A). The N50 and N90 lengths were 2.96 and 1.35 Mb, respectively (Table 1). A total of 94.59% of the 758 BUSCO genes and 96.77% of the 248 core genes by CEGMA were completely detected in the genome, indicating the completeness of the assembled genome (Appendix A). The previous assembly of *S. rugosoannulata* strain MG69 had 17,725 contigs with an N50 of only 8.32 kbp, and BUSCO was only 87.60% (Appendix A) [20].

### 3.2. Gene Prediction and Genome-Wide Functional Annotation

The prediction of TEs and other repetitive DNA sequences for the *S. rugosoannulata* genome identified that these regions comprised approximately 9.62 Mb or 19.91% of the genome, with TE accounting for 17.48% (Table 2 and Figure 2). Among the 16 tested species of Agaricomycetes, the proportion of repetitive sequences in the *S. rugosoannulata* genome was higher than most of the species (Figure 3).

The most abundant transposable and repetitive element types were Class I long terminal repeats (LTRs) with 5.39 Mb (11.15%), Class II DNA transposon with 1.75 Mb (3.61%), and long interspersed nuclear element (LINE) with 0.79 Mb (1.64%). Approximately 0.47% of the *S. rugosoannulata* genome was identified as tandem repeats, and a total of 22,553 SSRs were identified (Table 2).

To increase the accuracy, we used multiple tools to predict the gene structure and function (Appendix A). The final integration led to a dataset containing 11,750 protein-coding gene models with an average gene sequence length of 1942 bp, giving rise to a gene density of 47.21% in the assembled 48,331,048 bp. Among the 11,750 gene models, the average CDS length was 1455.07 bp, with an average exon number of 6.25 and 232.84 bp (Table 1).

For the non-coding gene, the rRNA cluster was found on scaffold 8 of the assembly, and the genes of 5.8S-18S-28S clustered in a region of 19 kb. There were 197 predicted tRNA genes, corresponding to 0.036% of the genome assembly length.

A total of 11,377 genes (96.83%) were annotated using the functional databases (GO, KEGG, KOG, SwissProt, and Nr) (Appendix A). According to the GO database, 5496 annotated genes were assigned to GO categories, with the first five as “metabolic process”, “binding”, “catalytic activity”, “cellular process”, and “single-organism process” (Appendix A). By mapping to the KEGG database, “Global and overview maps” accounted for the majority of the KEGG annotations, with 994 (25.98%) proteins classified into these categories (Appendix A). Other highly represented pathways were “signal transduction” with 330 (8.63%) and “transport and catabolism” with 320 (8.36%).

A total of 872 secreted proteins were predicted by assigning signal peptides by SignalP (http://www.cbs.dtu.dk/services/SignalP/. 7 October 2020) and removing transmembrane proteins with TMHMM. GO analysis showed that the secreted proteins were involved in hydrolase activity (*n* = 207, GO:0016787), the extracellular region (*n* = 41, GO:0005576), the cell wall (*n* = 41, GO:0005618), and the external encapsulating structure (*n* = 41, GO:0030312) (Appendix A).

### 3.3. Phylogenomic Analyses

The whole-genome sequences of 15 species of Agaricales were used for phylogenomic analysis (Appendix A). The species of *G. frondosa* (Polyporales) was included as an outgroup. The clustering of proteomes resulted in 8341 groups in the *S. rugosoannulata* genome, of which 1609 were core single-copy orthologous genes among 15 fungi, and 377 were unique gene families in *S. rugosoannulata* (Appendix A).

A maximum likelihood (ML) phylogeny analysis for *S. rugosoannulata* and the 15 additional fungal species was performed based on 1609 shared single-copy orthologous genes, which were concatenated into a supermatrix with 584, 537 amino acid sites (Figure 2a). Three clades of Agaricales, Agaricoid, Tricholomatoid, and Marasmioid [42] were significantly supported with high bootstrap values. *S. rugosoannulata* is phylogenetically close to *H. sublateritium* and *P. highlandensis* (Figure 2a). They all belong to Strophariaceae. The whole-genome sequences of *S. rugosoannulata* CGMCC 5.2220 and *H. sublateritium* FD-334 SS-4 showed a high level of sequence collinearity (Appendix A).

### 3.4. Putative Peroxidase-Encoding Genes in the Genome of S. rugosoannulata

Peroxidases can be divided into two significant groups contingent upon a heme cofactor’s presence or absence [38]. The *S. rugosoannulata* genome contains 50 heme peroxidases, 10 non-heme peroxidases, and one NADPH oxidase regulator (Appendix A). Putative peroxidase-encoding genes of all 63 Agricomycotina fungi recorded in fPoxDB (accessed on 20 December 2020), including biotrophs (ectomycorrhizal, mycoparasite, and root endophyte), BR and WR fungi, were retrieved and compared (Appendix A). There was an average of 34.7 peroxidase-encoding genes in the genome of these Agaricomycete fungi. Sixty-one peroxidase-encoding genes were identified in *S. rugosoannulata*, much more than the majority of the tested Agaricomycete fungi. Among the 61 peroxidase-encoding genes, there were 50 genes encoding heme peroxidases in *S.*
*rugosoannulata*, twice the average (25) of all of the tested fungi.

Phylogenetic trees were constructed based on the predicted amino acid sequences of the heme peroxidases and non-heme peroxidases of *S. rugosoannulata*, respectively, and the transcription levels were analyzed (Figure 4A,B). Among the 50 heme peroxidases, the proteins belonging to the same family generally were grouped together, including manganese peroxidase, hybrid ascorbate-cytochrome c peroxidases (also named hybrid-type A peroxidases, APx–CcPs), heme-thiolate peroxidases (HTPs), linoleate diol synthase, and NADPH oxidase genes (Nox). The relationships between peroxidase families were relatively weak, as indicated by the low bootstrap values (Figure 4A).

Four main groups of fungal class II peroxidases were classified by Floudas et al. [44] based on structure-functional properties: lignin peroxidase (LiP; EC 1.11.1.14), manganese peroxidase (MnP; EC 1.11.1.13), versatile peroxidase (VP; EC 1.11.1.16), and generic peroxidases (GP; EC 1.11.1.7). Seventeen putative MnP-encoding genes were identified in the genome of *S. rugosoannulata* and there were no genes encoding GP, LiP and VP. MnP proteins are defined as possessing an Mn (II)-oxidation site near the internal propionate of heme formed by three acidic residues referred to as *Phanerochaete chrysosporium* MnP1 Glu35, Glu39, and Asp179 and *Pleurotus eryngii* VPL Glu36, Glu40, and Asp175 [44]. The average number of MnP-encoding genes per species was only five in the 62 tested Agaricomycete fungi; however, there were 17 MnP-encoding genes in the genome of *S. rugosoannulata*. All 17 MnP proteins were atypical, with only two acidic residues at the Mn-oxidation site (Appendix A). It has been reported that 14 MnP genes in the genome of *H. sublateritium* are atypical [28]. The majority of MnPs were secreted proteins, and 13 of the 17 MnP genes were expressed in the mycelia of *S. rugosoannulata* (Appendix A and Figure 4A).

APx–CcPs share the enzymatic and structural features of ascorbate and cytochrome c peroxidases. Almost no Apx–CcP was found in most of the ectomycorrhizal and BR fungi. There was an average of one gene encoding APx–CcP in the genome of 62 tested Agaricomycete fungi (Appendix A); however, 10 were detected in the genome of *S. rugosoannulata*. *H. sublateritium*, the species in the same family of *S. rugosoannulata*, had 7 APx–CcP genes. Ten and eight genes were detected in the species *Galerina marginata* and *Gymnopus luxurians,* respectively. We then performed a phylogenetic analysis of the APx–CcP proteins from these four species. Generally, APx–CcP proteins from *G. marginata* and *G. luxurians* were clustered as a group, and those from *S. rugosoannulata* and *H. sublateritium* were clustered together (Figure 4C). All 10 APx–CcPs were secreted proteins, and SRUG_02817 showed the top three highest expression levels among all the peroxidase-encoding genes (Appendix A and Figure 4A).

Fourteen genes encoding heme-thiolate peroxidase (HTP) were identified in the genome of *S. rugosoannulata*. They all belong to haloperoxidase (chloroperoxidase, EC 1.11.1.10)-type sequences. HTP was present in all of the tested Agaricomycete samples except *Tremella mesenterica* and *Sebacina vermifera*, with an average of 7 copies per species. Several species showed expansions in HTP genes, including *A. bisporus* (24), *G. marginata* (24), *Auricularia subglabra* (16), *Exidia glandulosa* (30), *Fibulorhizoctonia* sp. (35), and *Sistotremastrum suecicum* (33). Most of them were transcripted in the *S. rugosoannulata* mycelia cultured in PDA medium.

Dye-decolorizing peroxidases (DyPs; EC 1.11.1.19) are a newly discovered family of heme peroxidases unrelated to well-known peroxidases in terms of their amino acid sequences, tertiary structures, and catalytic residues [45]. One gene encoding DyP was identified in the genome of *S. rugosoannulata*. DyP-encoding genes are widespread among WR genomes but are absent from the majority of BR genomes (Appendix A).

### 3.5. The Nutritional Strategy of S. rugosoannulata

A total of 371 genes could be assigned to CAZyme families, as defined in the CAZy database (Appendix A), which consisted of 165 genes encoding glycoside hydrolases (GHs), 107 auxiliary activity families (AAs), 55 glycosyltransferases (GTs), 25 carbohydrate esterase (CEs), 10 polysaccharide lyase (PLs), and 9 carbohydrate-binding modules (CBMs). Most of the Agaricales fungi that we have tested followed similar trends.

The terms WR, BR, and LD have traditionally been used to separate saprotrophic mushroom-forming fungi based on their nutritional strategies. Comparison of the genes associated with PCWD has revealed that LDs are significantly different from BR fungi, whereas LDs and WR fungi show both similarities and differences with respect to the composition of PCWD genes [25]. We then compared the genes encoding PCWD in *S. rugosoannulata* with LD, WR, and BR fungi. According to the previous research of Floudas et al., the genes encoding PCWD were divided into three functional groups: gene families encoding the enzymatic degradation of cellulose (12 families), hemicellulose (including pectin; 23 families), and lignin/xenobiotic (9 families) (Appendix A). For *S. rugosoannulata,* BR can be ruled out by a preliminary comparison of PCWD-encoding genes (Appendix A). It is difficult to determine whether *S. rugosoannulata* is an LD or WR fungus by direct analysis. Therefore, the nutritional strategy was analyzed by LDA analysis based on the 44 PCWD enzymes. Thirty-seven species from three groups (BR, LD, and WR) were examined. Ten characters with very low tolerance were excluded from the Failing Tolerance Test (default tolerance level 0.001) (Appendix A). The remaining 34 of the 44 characters were included in the analysis and grouped correctly at a rate of 100% (Appendix A). *S. rugosoannulata* was classified as an LD with 100% probability (Appendix A).

### 3.6. Detection of Secondary Metabolite Clusters

A total of 34 gene clusters located on different scaffolds were predicted using Antismash, including 1 nonribosomal peptide synthase (NRPS) and 11 NRPS-like gene clusters, 1 type 1 polyketide synthase (T1PKS), 16 terpene synthases (TSs), 2 siderophores, and 3 indole clusters (Appendix A). The number of secondary metabolite clusters in the genome of *S. rugosoannulata* was much greater than that of commercially cultivated mushrooms *A. bisporus*, *Lentinula edodes*, and *P. eryngii*, which suggests that diverse secondary metabolites can be produced by *S. rugosoannulata* (Appendix A).

Most fungi have no or only one siderophore cluster (Appendix A); however, two siderophore clusters have been identified in the genome of *S. rugosoannulata* (Figure 5A). Further analysis found that two core biosynthetic genes encoding siderophores in the majority of fungi, such as *H. sublateritium* and *Coprinopsis cinerea,* were adjacent and presented as a cluster (Figure 5B). In *S. rugosoannulata*, the related orthologous core biosynthetic genes for siderophores are located on scaffolds 2 and 4, so two siderophore clusters were identified (Figure 5B). Transcriptome analysis showed that all of the genes in the two siderophore clusters were expressed in the mycelia.

It was recently confirmed that the NRPS gene cluster is responsible for producing coprinoferrin in the Basidiomycete *C. cinerea* [46]. This gene cluster includes 17 genes in the genome of *C. cinerea,* and it was found that the 15 genes of the NRPS cluster (cluster 6 in Appendix A) in *S. rugosoannulata* shared a sequence similarity of over 40% to 80% with those of *C. cinerea* (Figure 5C and Appendix A) and *H. sublateritium*, respectively, and therefore might be responsible for the production of coprinoferrin-related compounds. Transcriptome analysis showed that all of the genes in the cluster were highly or moderately expressed.

Three indole clusters were found in the genome of *S. rugosoannulata*, far more than the majority of fungi that have no or only one indole cluster (Appendix A). However, the three core genes (SRUG_09987, SRUG_10137, and SRUG_11196) had no or very low expression in the mycelial sample.

### 3.7. No Genes Encoding Psilocybin Biosynthesis Enzymes Were Predicted in the Genome of S. rugosoannulata

The species of the genera *Conocybe*, *Gymnopilus*, *Panaeolus*, *Pluteus*, *Psilocybe*, and *Stropharia* have been reported to be hallucinogenic mushrooms that contain psilocybin [47,48]. Among the species of the genus *Stropharia*, *S. coronilla* has been reported to produce psilocin/psilocybin [49], and no psilocin/psilocybin was detected in *S. rugosoannulata* by high-performance liquid chromatography (HPLC) analysis [50]. Enzymatic synthesis of psilocybin has been reported in the phylogenetically closed species *P. cubensis* and *P. cyanescens* [29]. We searched the genes encoding three key psilocybin biosynthesis enzymes, namely, tryptophan decarboxylase (PsiD), psilocybin-related N-methyltransferase (PsiM), and psilocybin-related phosphotransferase (PsiK), in the genome of *S.*
*rugosoannulata*. It was found that the proteins SRUG_10179, SRUG_02421, and SRUG_10672 have only 38.5%, 27.5%, and 44.4% identity with the related enzymes responsible for psilocybin biosynthesis in *P. cubensis,* respectively. Phylogenetic analysis showed that the known psilocybin biosynthesis enzymes were grouped as a cluster with high support values, and the related proteins of *S.*
*rugosoannulata* are grouped into different clusters with PsiD, PsiK, and PsiM, which were confirmed to encode for psilocybin biosynthesis enzymes (Figure 6). This suggests that the gene cluster for psilocybin biosynthesis is absent in *S.*
*rugosoannulata*, which is consistent with the result of no psilocin/psilocybin determined by HPLC [50].

### 3.8. Gene Encoding Cytochrome P450 in the Genome of S. rugosoannulata

Blastp analysis was performed against the Fungal Cytochrome P450 Database (FCPD) and it was found that 217 cytochrome P450 monooxygenases in the *S. rugosoannulata* genome were classified into 57 families. The largest number was for CYP559 (26), followed by CYP65 (10) and CYP505 (9). Both CYP559 and CYP65 are involved in the secondary metabolism. There were 26 CYP-encoding genes involved in xenobiotic metabolism (Appendix A), which may participate in the diverse functions of detoxifying xenobiotic compounds. For example, SRUG_05169 (CYP504) encodes phenylacetate-oxidizing cytochrome P450 (EC 1.14.13), which can catalyze phenylacetate catabolism [51] and thus contributes to pollutant degradation.

## 4. Discussion

In this study, we presented the genome sequence of *S. rugosoannulata* generated by Illumina and PacBio RSII long-read sequencing technologies. The monokaryotic strain used in this study was obtained from the cultivar CGMCC 5.2220, which is widely cultivated in China. We focused on analysis of the repertoire of PCWD enzymes, nutritional strategies, and secondary metabolite clusters. *S. rugosoannulata* was classified as LD with 100% probability by LDA analysis based on PCWD enzymes. The expansion of genes encoding lignin and xenobiotic degradation enzymes, cytochrome P450 involved in the xenobiotic metabolism, and siderophore clusters confirm the potential application for bioremediation. A nutritional strategy of LD will guide the substrate selection for fruiting body cultivation.

The assembly of N50 2.96 Mb and N90 1.35 Mb was significantly improved from the previously reported version of this species, which had an N50 of 8,626 bp [20] (Appendix A). Since heterokaryotic stages are dominant during the lifecycle in the majority of basidiomycetes species, the monokaryotic strain helped obtain a high-quality genome. Currently, the high-quality genomes of some mushrooms, such as *Gloeostereum incarnatum* [52], *Pleurotus tuoliensis* [53], and *Sanghuangporus sanghuang* [54], have been sequenced and assembled with monokaryotic strains.

Peroxidases are a group of oxidoreductases that mediate the electron transfer from hydrogen peroxide (H_2_O_2_) or organic peroxide to various electron acceptors. The results of sequence analysis indicated that the number of heme peroxidase-encoding genes (50) in the genome of *S. rugosoannulata* was twice the average of all of the tested fungi. The distinctive features are the much greater HTP (14), MnP (17), and APx–CcP (10) than the majority of the tested Agaricales.

As a critical contributor to the microbial ligninolytic system, MnP can oxidize Mn^2+^ to oxidative Mn^3+^, which acts as a mediator for the oxidation of various phenolic compounds, as has been observed in the case of lignin or analogous structures such as xenobiotic compounds [55]. Atypical MnP has only one or two conserved acidic amino acid residues at the predicted Mn^2+^ binding site and has been identified in some wood-degrading fungi [44]. All 17 MnPs in *S. rugosoannulata* are atypical. HTPs are peroxidases that mediate the oxidation of halides and other compounds, including N-dealkylation, sulfoxidation, epoxidation of alkenes, and benzylic hydroxylation [56]. APx–CcP is a functional hybrid between cytochrome c peroxidase and ascorbate peroxidase. All of these peroxidase-encoding genes in the genome of *S. rugosoannulata* contribute to its potential application in bioremediation.

Three functional groups of gene families encoding PCWD enzymes (cellulose, hemicellulose, pectin, and lignin/xenobiotic) in the genome of *S. rugosoannulata* were compared to LD, WR, and BR. The ratio of total lignin and xenobiotic-related genes (50.5%) was higher than that of WR (39.6%), BR (29.7%), and LD (41.1%, Appendix A). The expansion of lignin- and xenobiotic-related genes in the genome of *S. rugosoannulata* is responsible for its potential application in bioremediation and its stronger ability to degrade lignin than most LDs.

*S. rugosoannulata* is found in forests or lawns on forest boards, fallen leaves, and rarely in decomposed wood [57]. Cultivation substrates include both cellulosic straw and lignin sawdust. Most studies have considered it an LD [9,16]; however, some reports have classified it as a WR [17,18]. It has been reported that LD fungi shares with WR fungi the plesiomorphic enzymatic network involved in cellulose decomposition, whereas genomic signatures related to hemicellulose- and lignin-degradation genes can separate LD fungi from most WR fungi [25]. The key differences are related to the absence of high-ligninolytic potential VP and LiP in most LD fungi compared to WR fungi. There were no VP or LiP genes in the genome of *S. rugosoannulata* (Appendix A), which is similar to that observed in most LD fungi. However, the number of genes encoding manganese peroxidase (17) was considerably higher than the average in LD (4) and WR (12) fungi. Enzyme activity analysis has confirmed that manganese-oxidizing peroxidase is the predominant enzyme of lignin degradation during growth in beech wood microcosms of *S. rugosoannulata* [13]. Additionally, GH11 genes were found only in LD and a few wood decayers. The average number of GH11 genes was four in LD and one in WR; however, there was only one GH11 gene in *S. rugosoannulata*. It seems to be challenging to classify *S. rugosoannulata* into LD or WR. Therefore, the nutritional strategy of *S. rugosoannulata* was analyzed by LDA based on the 44 PCWD enzymes, and *S. rugosoannulata* was classified as an LD with 100% probability (Appendix A).

The analyses of PCWD enzymes and nutritional strategy are important for substrate selection during fruiting body cultivation. For LD fungi such as *Agaricus bisporus*, composted cereal straw and animal manure are the key substrates for fruiting body cultivation [58]. However, sawdust is often used as one of the substrates and generally accounts for 30% of the medium for fruiting body cultivation of *S. rugosoannulata* in China. This is consistent with the expansion of genes encoding manganese peroxidase in the genome, which can help degrade lignin. On the contrary, as an LD fungus, *S. rugosoannulata* cannot grow well without cellulose. For example, when 100% sawdust was used to cultivate *S. rugosoannulata*, the spawn colonized on the substrates slowly, and there was almost no primordium differentiation or fruiting body formation (Figure 7), resulting in serious economic losses, whereas 100% bagasse fiber was good for fruiting body cultivation (Figure 7). As an LD fungus, *S. rugosoannulata* has a significant impact on the carbon cycle in terrestrial ecosystems as a saprotrophic decayer of leaf litter and straw.

Siderophores are generally synthesized by two pathways: An NRPS-dependent pathway and an NRPS-independent pathway. It has been confirmed that *CC1G_04210* (*cpf1*) encodes a siderophore synthetase and this NPRS cluster is responsible for the production of coprinoferrin, which is necessary for extracellular iron acquisition and is crucial for the growth and maturation of *C. cinerea* [46]. Homologous comparison and transcriptome analysis revealed that the NRPS cluster (cluster 6 in Appendix A) in *S. rugosoannulata* might be responsible for the production of coprinoferrin-related compounds. These NPRS siderophore synthetases are widespread in mushrooms and evolved from a common ancestor of basidiomycetes [46]. Moreover, two other siderophore clusters (clusters 4 and 7 in Appendix A) were predicted by Antismash in the genome of *S. rugosoannulata*. Pfam analysis showed that the core genes *SRUG_02402* and *SRUG_04293* have pfam04183 (IucA/IucC family) and pfam06276 (ferric iron reductase FhuF-like transporter). IucA and IucC are members of a family of non-NPRS-independent siderophore synthetases which are involved in the production of siderophores [59]. These two core genes encoding for siderophores, namely, *SRUG_02402* and *SRUG_04293*, are located on scaffolds 2 and 4 of the genome assembly of *S. rugosoannulata*, respectively, and two siderophore clusters were identified. However, the homologous genes of *SRUG_02402* and *SRUG_04293* in the other tested Agaricales mushrooms, such as *H. sublateritium* and *C. cinerea*, were adjacent and presented as a cluster (Figure 5A,B). Siderophores can bind to various metals, including iron. In addition to their essential role in the growth of fungi, siderophores have shown their potential roles in bioremediation [60]. Gene recombination might occur, and two clusters for the NRPS-independent siderophore synthesis were formed. It seemed that *S. rugosoannulata* can produce more siderophores for iron acquisition.

Psilocybin, a serotonin receptor agonist that induces altered states of consciousness, can be produced by some Agaricales, such as *Psilocybe cubensis*, *Gymnopilus dilepis*, and *Panaeolus cyanescens* [61]. *S.*
*rugosoannulata* has been widely cultivated in China and is recommended by the Food and Agriculture Organization as a cultivated mushroom for developing countries [2], implying safety. No psilocin/psilocybin was detected by HPLC analysis in *S. rugosoannulata* [50]. However, the genome data allowed us to make the first comprehensive inventory of the genes involved in psilocybin biosynthesis for comparison with known hallucinogenic mushrooms. Phylogenetic analysis revealed that the related proteins of *S.*
*rugosoannulata* were grouped into different clusters compared to the known proteins responsible for psilocybin biosynthesis (Figure 6). Consistent with safe usage as an edible mushroom, the *S. rugosoannulata* genome does not contain genes for known psilocin/psilocybin biosynthesis.

## 5. Conclusions

A high-quality genome of the *S. rugosoannulata* commercial cultivar in China was assembled, and the PCWD enzymes, nutritional strategy, and secondary metabolites were analyzed in this study. The expansion of genes encoding MnP, lignin and xenobiotic degradation enzymes, and cytochrome P450 involved in the xenobiotic metabolism in the genome of *S. rugosoannulata* can explain its strong ability to bioremediate and degrade lignin. Consistent with the fact that *S. rugosoannulata* is safe for use as an edible mushroom, the *S. rugosoannulata* genome does not contain genes for known psilocybin biosynthesis. Genome analysis of *S. rugosoannulata* will contribute greatly to fruiting body cultivation and will provide insight into its application in bioremediation.

## Figures and Tables

**Figure 1 jof-08-00162-f001:**
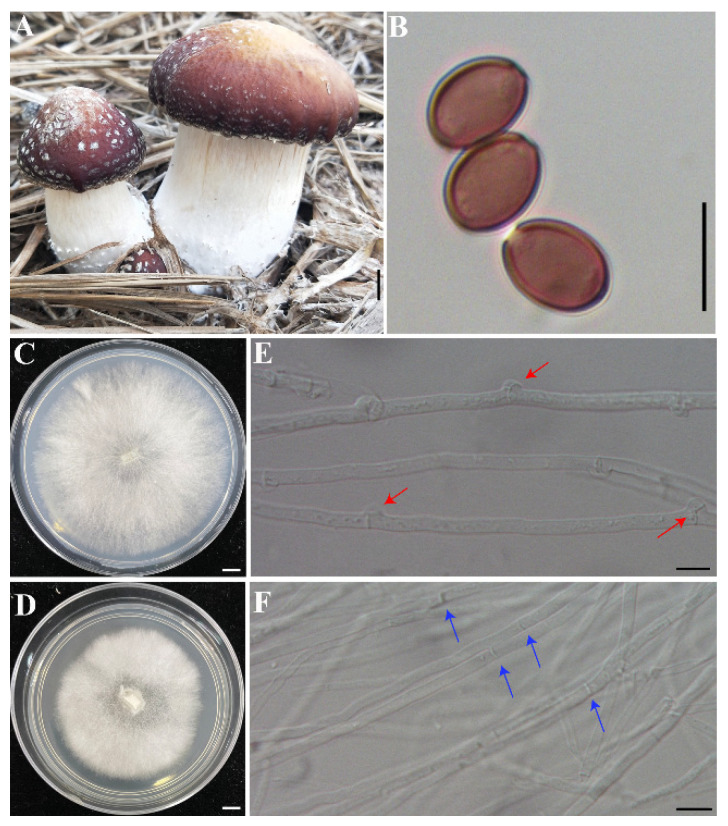
The fruiting bodies of *Stropharia rugosoannulata* and the monokaryotic strain used for genome sequencing. (**A**) The fruiting bodies of *S. rugosoannulata*. (**B**) The basidiospores of *S. rugosoannulata*. (**C**,**E**) Heterokaryotic mycelia with clamp connections. (**D**,**F**) Vegetative mycelium of monokaryotic strain S68. Bars: (**A**,**C**,**D**) = 1 cm; (**B**,**E**,**F**) = 10 μm.

**Figure 2 jof-08-00162-f002:**
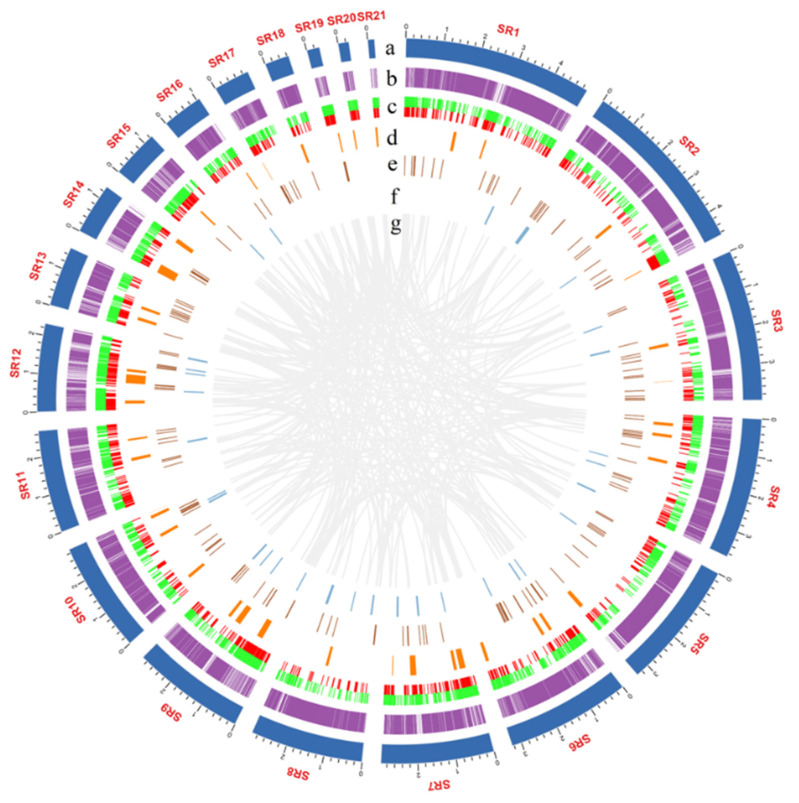
Circos graph of the characteristics of the *Stropharia rugosoannulata* genome. (**a**) 21 scaffolds. (**b**) Gene density. (**c**) Transposable elements (TEs), of which transposons are marked in red and retrotransposons in green. (**d**) SNP density. (**e**) Non-coding RNA. (**f**) Gene clusters of secondary metabolites. (**g**) Large fragment duplication.

**Figure 3 jof-08-00162-f003:**
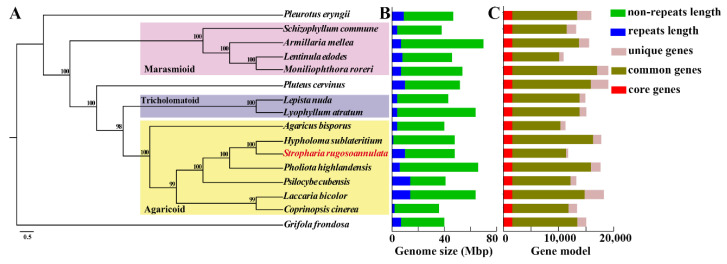
Phylogenetic tree and comparison of the genome features of *Stropharia rugosoannulata* with the other Agaricomycetes genomes. (**A**) Maximum likelihood tree showing the phylogenetic relationship based on 1609 single-copy orthologs of 16 Agaricomycetes fungi. (**B**) Genome size in Mbp, showing the distribution of the repeat and non-repeat contents. (**C**) Gene model counts of each genome are divided into core genes (present in all genomes), common genes (present in two or more genomes), and unique genes (exclusively found on that genome).

**Figure 4 jof-08-00162-f004:**
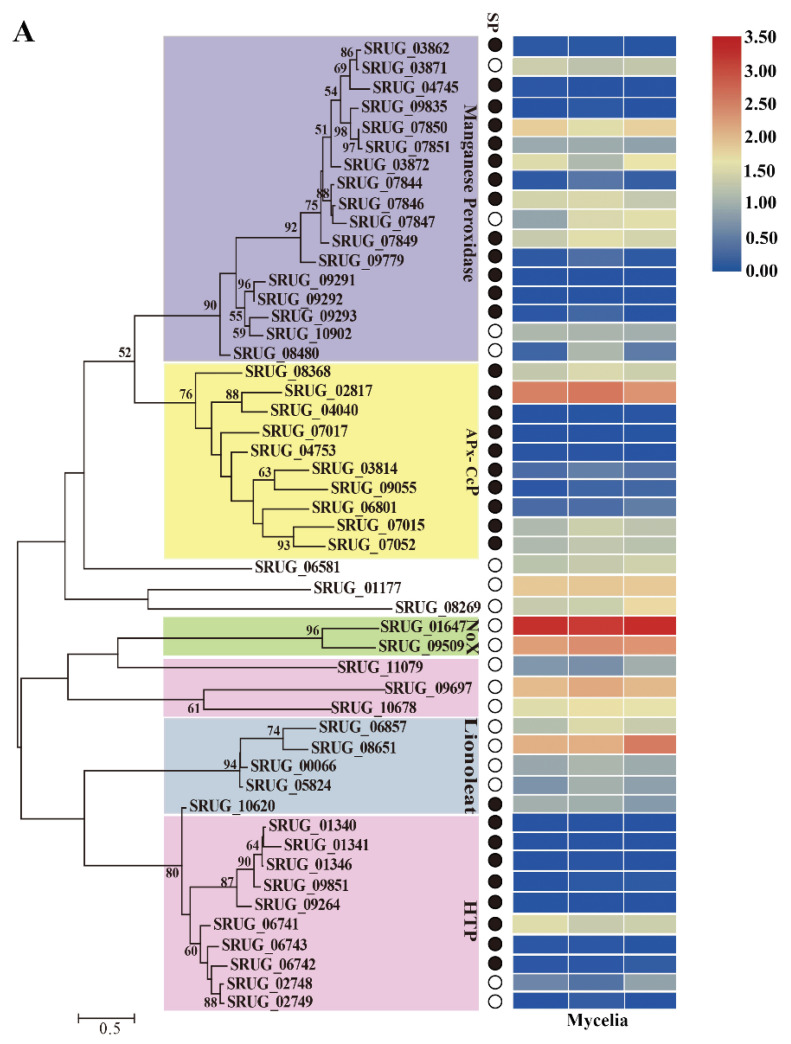
Phylogenetic analysis based on the amino acid sequences of peroxidases and transcription level of the genes (**A**) 50 putative heme peroxidases. (**B**) 10 putative non-heme peroxidases. (**C**) Phylogenetic analysis of Apx–CcPs from the species of *S. rugosoannulata*, *H. sublateritium*, *G. marginata,* and *G. luxurians*. The phylogenetic trees were constructed using RAxML (version 8) with 1000 bootstrap replicates. The numbers at the nodes represent the bootstrap percentages. The solid black and empty circles represent the secreted and non-secreted proteins, separately. A heatmap drawn with TBtools [43] indicated the expression in the mycelia of *S. rugosoannulata*. HTP, heme-thiolate peroxidases, APx–Ccp, hybrid ascorbate-cytochrome c peroxidases, and Nox, NADPH oxidase genes. The red fonts in (**C**) indicated the Apx–CcPs from *S. rugosoannulata*.

**Figure 5 jof-08-00162-f005:**
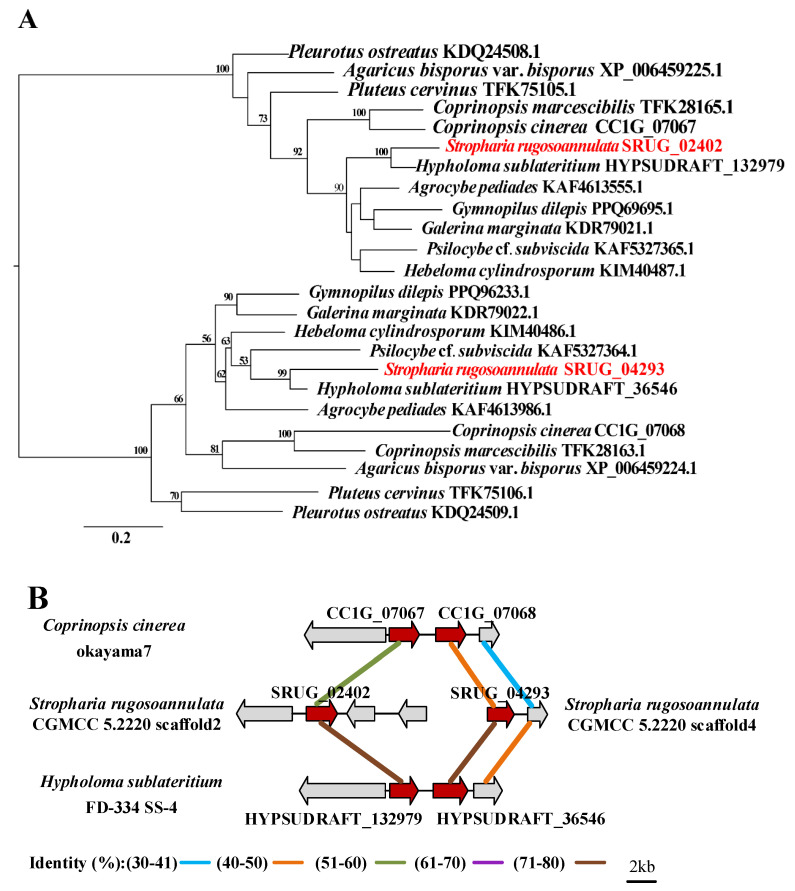
Comparative analysis of the siderophore gene clusters. (**A**) Phylogenetic analysis based on the amino acid sequences of the core genes of siderophore clusters. The red fonts in Figure 5A indicated the proteins of *S. rugosoannulata*. (**B**) Synteny analysis of the siderophore-encoding gene cluster predicted by Antismash in *C. cinerea*, *S. rugosoannulata,* and *H. sublateritium*. (**C**) Synteny analysis of the gene cluster responsible for coprinoferrin biosynthesis in *C. cinerea*, *S. rugosoannulata,* and *H. sublateritium*. The identity is based on the protein sequences. Gray arrowheads indicate the homologous genes, while white arrowheads indicate unique genes within the region analyzed and crimson arrowheads indicate the NRPS core genes. The sequence identity between the homologous proteins from the two fungi is shown by bold solid lines of different colors.

**Figure 6 jof-08-00162-f006:**
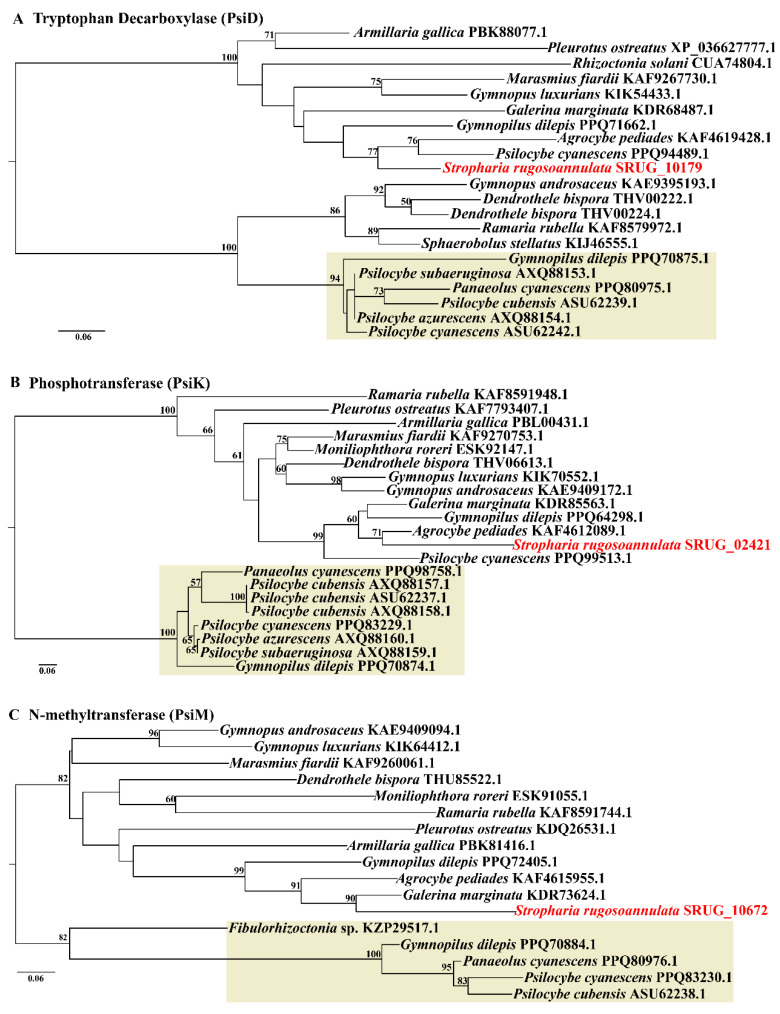
RAxML phylogenetic analysis of the key psilocybin biosynthesis enzymes. (**A**) Tryptophan decarboxylase (PsiD); (**B**) phosphotransferase (PsiK); (**C**) N-methyltransferase (PsiM). The known psilocybin biosynthesis enzymes were grouped as a cluster, which is indicated in pale yellow. The red fonts indicated proteins of *S. rugosoannulata*.

**Figure 7 jof-08-00162-f007:**
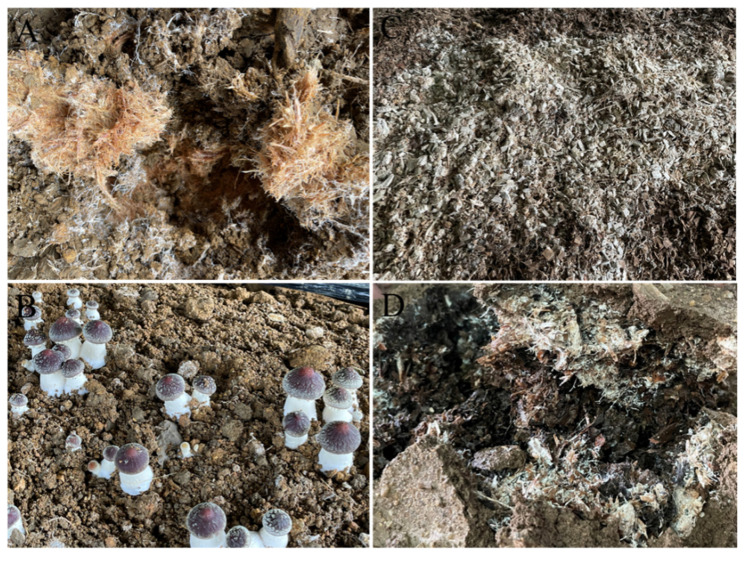
The mycelia and fruiting body of *Stropharia rugosoannulata* growing on bagasse fiber and sawdust. (**A**,**B**) The 100% bagasse fiber as substrate to cultivate *S. rugosoannulata*; (**C**,**D**) *S. rugosoannulata* mycelia growing on 100% sawdust.

**Table 1 jof-08-00162-t001:** Summary of the genome assembly and annotation of *Stropharia rugosoannulata*.

Genome	Value
Length of genome assembly, bp	48,331,048
No. of scaffolds	21
Length of the largest scaffold, bp	4,928,370
Length of the smallest scaffold, bp	182,917
N50, bp	2,961,130
N90, bp	1,348,662
Scaffolds ≥ 5 kb, percentage of assembly, %	100
GC content, %	47.35
No. of protein-coding genes	11,750
Average protein length, aa	485
Average exon size, bp	232.84
Average No. of exons per gene	6.25
Average intron size, bp	92.77

**Table 2 jof-08-00162-t002:** Classification of the repeat sequences in the genome of *Stropharia rugosoannulata*.

Classification	Order	Superfamily	Number of Elements	Length of Sequence (bp)	Percentage of Sequence (%)
Class I (retrotransposons)			7915	6,192,475	12.81
	LTR		5813	5,386,682	11.15
		Gypsy	2292	3,629,891	7.51
		Copia	987	521,946	1.08
		Unknown	2534	1,186,845	2.55
	LINE		1986	790,514	1.64
	SINE		116	15,279	0.03
Class II (DNA transposons)			3706	2,254,847	4.67
	DNA		2473	1,745,535	3.61
	MITE		1090	417,182	0.86
	RC	Helitron	143	92,130	0.19
Total TEs			11,621	8,447,322	17.48
Tandem Repeats			4087	227,073	0.47
	Ttandem repeat		2288	204,520	0.42
	SSR		1799	22,553	0.05
Simple repeats			203	19,120	0.04
Unknown			2463	927,192	1.92
Low complexity			9	1419	0.00
Total repeats			18,383	9,622,126	19.91

## Data Availability

Raw sequences of both PacBio long-read sequencing and Illumina short-read sequencing were submitted to NCBI SRA (http://www.ncbi.nlm.nih.gov/sra accessed on 3 January 2022) under BioProject PRJNA690158. Transcriptome data were submitted under BioProject GSE164538.

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
