# Peer review of "Genomic Analysis of Stropharia rugosoannulata Reveals Its Nutritional Strategy and Application Potential in Bioremediation"

_jof, 2022, doi:10.3390/jof8020162_

Round 1
Reviewer 1 Report
The manuscript ‘Genomic analysis of Stropharia rugosoannulata reveals degradation mechanisms on lignin and xenobiotic and the nutritional strategy’ corresponds to an updated version of the manuscript previously submitted to Journal of Fungi. From the methodical point of view I have no complaints. In this submission the authors have addressed few recommendations to the previous version of the manuscript. The language is slightly improved, albeit the same type of fundamental grammar errors are still present.
Again, the entire manuscript, especially introduction and results should be verified by person skilful in the rules of English grammar. Reviewer is not for correction of fundamental grammatical errors. I don't have the strength anymore to correct fundamental grammatical errors, hence, only some rows are listed here, required to rewritten, to be readable.
Title is still illogical. Should be rewritten to indicate the main subject.
- sentence order and style, required to be rewritten, row: 131, 150-151, 281-keep those?, 359-360-only less?, 381, 422-423, 453, 493, 511, 513, 651, 770, 780-783, 820-pollution degradation, 836, 912, 958, 986, 996, 1002, 1007, 1009
- again, lack of predication or subject, row 154, 451, 472, 662, 784
- wrong choice of synonyms, row: 831
Row 834: past tense ‘helped’-sequencing was done
Preposition, row: 721, 912
constant confusion of gene with protein, row: 30-32, 487, 510, 515, 658-659, 700, 750, 812, 827, 997, 1024-1025
Lack information how the libraries for transcriptomic analysis have been done.
Row 514: the average of what? Length? Localization?
Figure 5A: phylogenetic analysis of the DNA sequence or encoded proteins?
Row 813: blastp has been conducted by itself? Or it is a type of bioinformatic analysis done by author?
Row 984-985: a NRPS
Row 1017: which proteins? Related? Homologous?
Author Response
The reply to reviewer1 is a word file.

Reviewer 2 Report
In my opinion, the Authors improved the manuscript entitled "Genomic analysis reveals the potential in bioremediation and nutritional strategy of Stropharia rugosoannulata", according to the reviewer comments. In my opinion, the article deserves publication.
Author Response
Thanks for the positive comments and hard work on our Ms.
Round 2
Reviewer 1 Report
The manuscript ‘Genomic analysis of Stropharia rugosoannulata reveals the nutritional strategy and application potential in bioremediation’ corresponds to an updated version of a manuscript previously submitted to Journal of Fungi. This time, the language has been improved and the manuscript is comprehensible. I have only small remarks:
In the title consider use genitive: ...reveals its nutritional... to indicate what was sequenced and whose nutritional strategy is described in text
row 30: fungus, based (comma separated)
row 135: five fungi: Hypholoma...(colon)
row 361: parentheses are separated by comma
Figure 5A: style - the sentence is incomplete. It is an equivalent of sentence
Author Response
Response to reviewer 1
Comments and Suggestions for Authors
The manuscript ‘Genomic analysis of Stropharia rugosoannulata reveals the nutritional strategy and application potential in bioremediation’ corresponds to an updated version of a manuscript previously submitted to Journal of Fungi. This time, the language has been improved and the manuscript is comprehensible. I have only small remarks:
R: Thanks very much for the work on our Ms. We have revised the Ms. carefully following the suggestion one by on.
In the title consider use genitive: ...reveals its nutritional... to indicate what was sequenced and whose nutritional strategy is described in text
R: Agreed and thanks. The title has been revised as “Genomic analysis of Stropharia rugosoannulata reveals its nutritional strategy and application potential in bioremediation”.
row 30: fungus, based (comma separated)
R: Agreed and added.
row 135: five fungi: Hypholoma...(colon)
R: Agreed and revised.
row 361: parentheses are separated by comma
R: Agreed and revised.
Figure 5A: style - the sentence is incomplete. It is an equivalent of sentence
R: “such as” has been added in line 375.
“Figure 5A Phylogenetic analysis based on the amino acid sequences of the core genes of siderophore clusters.” We think as the legend of the figure, it should be OK.
This manuscript is a resubmission of an earlier submission. The following is a list of the peer review reports and author responses from that submission.
Round 1
Reviewer 1 Report
Dear Authors,
The article titled "Genomic analysis of Stropharia rugosoannulata reveals degradation mechanisms on lignin and xenobiotic and the nutritional strategy" provided a lot of useful information about the genes involved in cellulose and lignin degradation. The genome analysis of Stropharia rugosoannulata allows a better understanding of many fungal processes in the future. In my opinion, the article deserves publication after a few improvements.
Title
In my opinion, the title "Genomic analysis of Stropharia rugosoannulata reveals degradation mechanisms on lignin and xenobiotic and the nutritional strategy" is not adequate for the analysis conducted in the research article. Only putative genes were presented in this study, and it is only the prediction of gene structures and functions. This study reveals only putative genes encoding enzymes involved in biodegradation. In the future, this study may be helpful in gene expression analysis and primers designed.
Abstract
The abstract is well written and has all the information described in the article.
The sentence 28-30, "Genome analysis of Stropharia rugosoannulata will enhance our understanding of cellulose and lignin degradation by this fungus, as both of these are critical processes in the global carbon cycle, and will provide insight into its application in bioremediation," should be rewritten. The article is described the predicted genes of S. rugosoannulata involved in degradation.
I don't understand why the Authors mentioned the application in bioremediation of S. rugosoannulata. S. rugosoannulata is known as ligninolytic fungi. So far, there are many papers about the degradation of different environmental pollutants. (e.g., Fungal bioremediation of diuron-contaminated waters: evaluation of its degradation and the effect of amendable factors on its removal in a trickle-bed reactor under nonsterile conditions, Hu et al. 2020; Stropharia rugosoannulata and Gymnopilus luteofolius: Promising fungal species for pharmaceutical biodegradation in contaminated water, Francesc Castellet-Rovira et al. 2018).
Introduction
43-48. In my opinion, these sentences are unnecessary in the Introduction because the pharmacological activities of the fungus Stropharia rugosoannulata do not correlate with the work and analysis done in this research paper. In my opinion, the Authors should focus on describing enzymes and nutritional strategy. Moreover, information from lines 53-61 is also unnecessary because the analysis described in this research paper focuses on predicted genes encoding enzymes involved in degradation. The authors should focus on degradation efficiency on different medium or cultivation conditions, enzymes activity, or describe the gene expression analysis.
(Patterns of lignin degradation and oxidative enzyme secretion by different wood- and litter-colonizing basidiomycetes and ascomycetes grown on beech-wood , Christiane Liers et al. 2011, Fungal bioremediation of diuron-contaminated waters: evaluation of its degradation and the effect of amendable factors on its removal in a trickle-bed reactor under non-sterile conditions, Kaidi Hu et al. 2020).
68 please replace "mushrooms" with "fungal species."
67/73 Please include research articles in the Introduction:
"Analyses of mating systems in Stropharia rugosoannulata based on genomic data" Jun-Jun Shang et al. 2020;
"The complete mitochondrial genome sequence of the edible mushroom Stropharia rugosoannulata (Strophariaceae, Basidiomycota)" Tomohiro Suzuki et al. 2018;
"Study of thermotolerant mechanism of Stropharia rugosoannulata under high temperature stress based on the transcriptome sequencing" Jifan Ren et al. 2021;
"Transcriptomic Analysis of Stropharia Rugosoannulata Reveals Potential Carbohydrate
Metabolism and Cold Resistance Mechanisms Under Low-Temperature Stress" Haibo Hao et al. 2021
373 "Phylogenetic analysis showed that S. rugosoannulata proteins grouped into different clusters compared to PsiD, PsiK, and PsiM for psilocybin biosynthesis enzymes (Figure 6). The analysis suggests that the gene cluster for psilocybin biosynthesis is absent in S. rugosoannulata, and it cannot produce psilocybin".
I cannot entirely agree with the authors' opinion because even the proteins grouped into different clusters do not mean that the psilocybin is not synthesized. Only the mass spectrometry analysis can confirm this thesis. And there is many research paper that ensures that S. rugosoannulata can produce psilocybin (e.g., Accidental death involving psilocin from ingesting "magic mushroom," Uttam Garg et al. 2020). Even the proteins have only 27.5-44.4% identity with the related enzymes responsible for psilocybin biosynthesis in P. cubensis and P. cyanescens. There is no confirmation that this compound is not produced. The differences in gene clusters of Psilocibe sp. and Stropharia rugosoannulata could have occurred due to the rapid evolutionary divergence of fungi and mutations in the area of this gene cluster.
Results
The data included in Fig. 4c are hardly readable. The authors are asked to improve the form of the presentation.
Author Response
I will submit word file that reply to reviewers.

Reviewer 2 Report
In the manuscript ‘Genomic analysis of Stropharia rugosoannulata reveals degradation mechanisms on lignin and xenobiotic and the nutritional strategy’, the authors sequenced and assembled genomic sequence of commonly cultivated mushroom Stropharia rugosoannulata, further performed detailed genomic analysis and functional annotation of encoded proteins. Based on the analysis of cell wall degrading enzymes, the authors concluded that S. rugosoannulata should be classified as a litter-decomposing fungus. Due to plethora of CAZymes produced by S. rugosoannulata, the fungus seems to be promising bioremediating organism.
In my opinion, the topic analyzed here, is of interest for the readers of Journal of Fungi as an important study of another sequenced basidiomycetous fungus, its genomic analysis of biochemical pathways toward its practical application in bioremediation. The high-throughput sequencing and detailed in silico analysis done here are impressive. From the methodical point of view I have no complaints. However, I have a major remark regarding this work: the entire manuscript, especially abstract, introduction and results must be rewritten, obeying the rules of English grammar. Language correction by the person fluent in English is required.
Some examples are presented below:
Title should be rewritten. In the present form is illogical without sense: 'reveals' is unsuitable in this case, 'nutritional strategy' is not degraded as the sentence order indicates. At most, 'degradation mechanisms' may be a part of 'nutritional strategy'.
- sentence order and style: row 43 (In recent years, the consumption...), row 362-Syntheny analysis of siderophore-encoding gene cluster..., row 324-326-...showed much more (of what?/than what?)...differences (in what?) were found...LD or WR,(which?) cannot...Row 413-414 to rewrite. Row 415-..which was (refers to S. rugosoannulata or LiP genes?), row 380- ..and ‘it’ cannot...refers to S. rugosoannulata or the gene cluster (row above)? Row 392-..to catalyze phenylacetate catabolism – it is completely unclear. Please, rewrite this curiosity.
- logical quantifier: row 425- hydrogen peroxide AND organic peroxide simultaneously?
- wrong tenses: row 23-genome..IS reported (in the current manuscript), row 391 (CYP504) encoded-and currently not?
- wrong choice of synonyms: in general: row 315, 337, 353, 404 majority, rather than most, row 282-functional properties.
- lack of predication or subject: row 267-generally (are?) clustered..., row 303-...H. sublateritium (are?) clustered.., row 326-LD or WR,(which?) cannot..., row 336-...there (where?) much (how much? rather more)..than..(-sentence order is wrong. Please, rewrite this), row 343-...for siderophore (are?) located..., 378-proteins (are?) grouped into different clusters (as?) compared..., row 417-...WR was (observed?) that..., row 426-...the genome was more or S. rugosoannulata was more (than what?), row 430-as (it is observed?) in the case.
- constant confusion of gene with protein: proteins or amino acids are NOT located in the genome: row 152 – peroxidase ENCODING genes..., row 315- DyP-encoding genes, row 341-genes (encoding) siderophore in the most (?how the most?), row 431-432-since when atypical genes possess amino acids? What is atypical in the genes?
- sentence off the topic: row 46 (what worm refers to genome sequencing?), row 465-470- paragraph of psylocibin is out of scope. Please, rewrite this.
And few minor remarks to improve the clarity of the manuscript, as listed below:
row 43: ‘mu’ is not a unit of SI metric system
row 69: please, expand the information about previous sequencing. In the current one, presented here, heterokaryotic isolate, presumably of one MAT sign, has been sequenced. have You sequenced and analyzed MAT loci in this species?
Row 77: how S. rugosoannulata genome sequence will contribute to food?
Row 82: ‘as a routine’ sounds awkward
Row 221: ‘falling’ sounds awkward, like slang. Better use ‘are classified’
Row 240: expand ML abbreviation
Rows 256-266, 426: please, declare, which form You use: heme (modern) or haem (ancient)
Row 287: referred, rather than homologous
Row 288: There was only an average – unclear phrase
Row 299: Ten and eight, or 10 and 8
Row 304: 3 high expression level
Row 308: expansion of HTP
Row 314: One gene (of what?) – the sentence is incomplete and illogical. Please, rewrite
Row 339: if in genome were identified two siderophore clusters, than their amount cannot be excepted by ‘but’
Row 348: homologous comparison sounds like tautology
Row 358: since the authors suggest indole clusters as involved in nemathophagous lifestyle, it would be mandatory to analyze secreted protease, lipase and chitinase repertoire
Row 402: monokaryotic... repetition
Row 403: what it means superior and how it looks?
Row 407: fungi were sequenced or they were a tool for sequencing? Who sequenced whom?
Row 410: some reports classified it
Row 414: compared to
Row 422-423 may be moved to Materials and Methods; it is technical description
Row 425: have You checked experimentally peroxidases? Or only in silico? Then, please write properly – peroxidase sequences analysis.
Row 430: oxidative Mn3+
Row 447: maturation of C. cinerea
Row 478: ability to bioremediate
Row 489: is that genomes are commercially available, or they are a property of JGI, required permission for use?
Author Response

(The authors gave the same response as above.)

Round 2
Reviewer 2 Report
In the manuscript ‘Genomic analysis of Stropharia rugosoannulata reveals degradation mechanisms on lignin and xenobiotic and the nutritional strategy’, the authors sequenced and assembled genomic sequence of commonly cultivated mushroom Stropharia rugosoannulata, further performed detailed genomic analysis and functional annotation of encoded proteins. This submission corresponds to an updated version of a manuscript previously submitted to Journal of Fungi. The authors have addressed a few, only indicated my recommendations to the previous version of the manuscript.
Again, the entire manuscript, especially abstract, introduction and results should be verified by person skilful in the rules of English grammar. I insist on correction by native speaker. Admitting a comprehensible text lies with the editor site and is not the job of the reviewer. Reviewer is not for correction of fundamental grammatical errors.
Title is still illogical and not corrected. 'Nutritional strategy' is not degraded as the sentence order indicates. At most, 'degradation mechanisms' may be a part of 'nutritional strategy'. Or, if it is listing of abilities, then ‘genomic analysis reveals’ should refers somehow to xenobiotic.
- sentence order and style: row 23- ‘is reported’ refers to genome, fungus or China?, row 34-the new sentence sounds awkward ‘enhance our understanding’-please, rewrite this curious missed synonyms, row 78-‘should be’?-style, row 82-‘characterized by bioinformatics based on genetic data’-order and style, row-85 ‘is urgently’-style, row 139-‘S. rugosoannulata was performed’?-order, row 391-style, row 412-style, illogical, row 456: ‘It’ refers to manganese or system? wrong order, row 459-460- order, row 484-‘of..of’ style and order, row 488- ‘in addition’? style, row 496-497-order
- again, lack of predication or subject: row 245-transposons are, row 390 lack of predication to enzymes
- wrong choice of synonyms: exempt
Row 27: again, ancient version of heme
Row 28: again, P450 encoding
Row 55: genitive (who, what) of water
Row 83: genitive (who, what) thermotolerance
Row 84: ‘was deficient’ of what? The sentence is incomplete
Row 93: ‘origins’ – thus, there are more than one origin of the sequenced strain?
Row 100: supposingly, monokaryotic strain S68 is derived of Heinong strain? It should be described
Row 174: is rather to Materials and Methods
Row 236: listing: (colon) ‘signal...’
Row 277: conjunction ‘than...most’
Row 279: conjunction ‘twice...the’
Row 454: conjunction ‘feature...(next genitive)’
Row 400: ‘44.4%, respectively’
Row 442-445: split into 2 sentences; ‘peroxidase was’ than cannot be plural ‘enzymes’. ‘Additionally’
Row 452: ‘sequence analysis’ cannot ‘found’ because analysis is not an autonomous person
Row 477: ‘were in charge’ ? what does it mean?
Row 491: the fungus has to clusters rather for the iron (and other metal ions) acquisition, than bioremediation. Bioremediation is a technique invented by human, not fungus
Row 501: altered genomic localization and/or different loci organization (lack of syntheny) does not exclude metabolite production, vide: ergot alkaloids production by Epichlöe, where gene cluster is spited and distributed in two chromosomes
Author Response
I will submit word file to reply to reviewer.
